# Association between Loss of Immune Checkpoint Programmed Cell Death Protein 1 and Active ANCA-Associated Renal Vasculitis

**DOI:** 10.3390/ijms24032975

**Published:** 2023-02-03

**Authors:** Samy Hakroush, Björn Tampe

**Affiliations:** 1Institute of Pathology, University Medical Center Göttingen, 37075 Göttingen, Germany; 2SYNLAB Pathology Hannover, SYNLAB Holding Germany, 86156 Augsburg, Germany; 3Department of Nephrology and Rheumatology, University Medical Center Göttingen, 37075 Göttingen, Germany

**Keywords:** renal vasculitis, programmed cell death protein 1, complement factor B

## Abstract

Immune checkpoint inhibitors (ICIs) have made an important contribution to the survival of patients with certain cancers. ICIs interrupt co-inhibitory signaling pathways mediated by programmed cell death protein 1 (PD-1), programmed cell death protein ligand 1 (PD-L1) and cytotoxic T lymphocyte-associated antigen (CTLA-4) that result in the elimination of cancer cells by stimulating the immune system. However, immune-related adverse events have also been described and attributed to an enhanced immune system activation. Recent observations have suggested a dysregulation of immune checkpoints in active antineutrophil cytoplasmic antibody (ANCA)-associated vasculitis (AAV). We here analyzed intrarenal PD-1 and PD-L1 by immunostaining in a total of 15 kidney biopsies with ANCA-associated renal vasculitis in correlation with glomerular and tubulointerstitial lesions. For independent validation, publicly available datasets were analyzed for PD-1 expression (encoded by *PDCD1*). We here observed a predominant tubulointerstitial expression of PD-1 that is decreased in ANCA-associated renal vasculitis. Moreover, loss of tubulointerstitial PD-1 correlated with active ANCA-associated renal vasculitis. Consistent with the observed association with active glomerular and tubulointerstitial lesions, we identified that interstitial PD-1 correlated with tubular and/or glomerular PD-L1 positivity. Finally, PD-1 was associated with decreased local synthesis of complement factor B. Interestingly, we did not observe a correlation between PD-1 and complement C5 or its C5a receptor. Combined with our observations, this may implicate a link between impaired PD-1/PD-L1 signaling, complement factor B and active ANCA-associated renal vasculitis. These findings could be of relevance because experimental data have already described that PD-1 agonism can be used therapeutically to attenuate autoimmunity in multiple disease models. Furthermore, targeted therapy against a complement C5/C5a receptor and factor B are both available and currently evolving in the treatment of AAV. Therefore, this pilot study expands our current knowledge and describes a potential interplay between immune checkpoints and the alternative complement pathway in active ANCA-associated renal vasculitis.

## 1. Introduction

Immune checkpoint inhibitors (ICIs) have made an important contribution to the survival of patients with certain cancers. ICIs interrupt co-inhibitory signaling pathways mediated by programmed cell death protein 1 (PD-1), programmed cell death protein ligand 1 (PD-L1) and cytotoxic T lymphocyte-associated antigen (CTLA-4) that result in the elimination of cancer cells by stimulating the immune system. However, immune-related adverse events have also been described and attributed to an enhanced immune system activation. Recent observations have suggested a dysregulation of immune checkpoints in active antineutrophil cytoplasmic antibody (ANCA)-associated vasculitis (AAV) [1]. This is in line with the fact that immune checkpoint molecules are present in injured kidneys independent of ICIs [2]. Regarding renal vasculitis, it was recently reported that specifically PD-1 inhibitors could cause de novo or relapsing ANCA-associated renal vasculitis [3]. These observations implicate the presence of the target molecule and a protective role of PD-1 signaling in renal vasculitis. AAV is a small vessel vasculitis, most frequently presenting as microscopic polyangiitis (MPA) or granulomatosis with polyangiitis (GPA) [4,5]. Acute kidney injury (AKI) is a common and severe complication of AAV as it can cause progressive chronic kidney disease (CKD), end-stage renal disease (ESRD) or death [6,7]. Pathogenic ANCAs, in particular proteinase 3 (PR3) and myeloperoxidase (MPO), trigger a deleterious immune response resulting in pauci-immune necrotizing and crescentic renal vasculitis, a common manifestation of glomerular injury in AAV [8]. Seminal work has shown that complement depletion by cobra venom factor and blockade of the alternative complement pathway protected from experimental renal vasculitis [9]. Contrasting to this, complement C4 knockdown as a shared component of the classical and lectin pathway had no effect [9]. These findings first demonstrated that the alternative complement pathway is a prerequisite of ANCA-induced lesions. Interestingly, interstitial PD-1 has already been described as a determinant of susceptibility to kidney injury in the context of ICIs and has been linked to regulation by the complement system [10]. However, the presence of PD-1 and its association with the complement system in renal vasculitis have not been described yet. We hereby aimed to analyze the abundance of immune checkpoint molecules PD-1/PD-L1 and its implications in ANCA-associated renal vasculitis.

## 2. Results

### 2.1. Immune Checkpoints PD-1 and PD-L1 Are Present in Different Renal Compartments in ANCA-Associated Renal Vasculitis

We first analyzed the abundance of PD-1 and PD-L1 among different intrarenal compartments in ANCA-associated renal vasculitis. In ANCA-associated renal vasculitis, immune checkpoint molecules PD-1 and PD-L1 were present (Figure 1A). While the presence of PD-1 was limited to interstitial cell positivity, PD-L1 was predominantly found in the tubular and glomerular compartment in ANCA-associated renal vasculitis (Figure 1B). In the case of PD-1 positivity, most were also positive for tubular and/or glomerular PD-L1 (Figure 1C). Interestingly, we did not observe tubular and/or glomerular PD-L1 positivity without the presence of PD-1 (Figure 1C).

### 2.2. Loss of Interstitial PD-1 Correlates with Active ANCA-Associated Renal Vasculitis

Correlative analysis revealed that the loss of PD-1 associated with kidney injury in ANCA-associated renal vasculitis was attributed to active lesions including glomerular crescents (*p* = 0.0061) and necrosis (*p* = 0.0013), more tubulointerstitial inflammation including tubulitis (*t*, *p* = 0.0093), total inflammation (*ti*, *p* = 0.002) and inflammation in areas of interstitial fibrosis/tubular atrophy (*i-IFTA*, *p* = 0.0086, Figure 2A). This observation was further supported by subgrouping according to the Berden classification and ANCA renal risk score (ARRS); active ANCA-associated renal vasculitis correlated with the loss of interstitial PD-1 (Figure 2B,C).

### 2.3. PD-1 Associates with Decreased Local Synthesis of Complement Factor B

For independent validation, we next extracted mRNA levels of *PDCD1* (encoding PD-1) from transcriptome array datasets. Here, we confirmed a predominant *PDCD1* mRNA expression within the tubulointerstitial as compared to the glomerular compartment (*p* < 0.0001, Figure 3A). Interestingly, *PDCD1* mRNA expression was significantly lower in ANCA-associated renal vasculitis as compared to healthy control kidneys (*p* = 0.0006, Figure 3B). Correlative analysis confirmed the association between loss of intrarenal *PDCD1* mRNA expression and increased serum creatinine measurements (*p* = 0.0005, Figure 3C). Among complement system components, multivariate analysis revealed the strongest correlation between intrarenal mRNA levels of *PDCD1* and decreased *CFB* (encoding complement factor B, *p* = 0.0001, Figure 3D).

## 3. Discussion

We here observed a predominant tubulointerstitial expression of PD-1 that is decreased in ANCA-associated renal vasculitis. Moreover, the loss of tubulointerstitial PD-1 correlated with active ANCA-associated renal vasculitis. Consistent with the observed association with active glomerular and tubulointerstitial lesions, we identified that interstitial PD-1 correlated with tubular and/or glomerular PD-L1 positivity. Experimental studies have already described that disruption of the tissue-protective PD-1/PD-L1 checkpoint unleashes immunity in the pathogenesis of medium and large vessel vasculitites [11]. T-cell-dependent immune responses are modulated by costimulatory and coinhibitory stimuli, particularly receptor–ligand interactions that modulate T-cell receptor (TCR) signaling [12]. Such immune checkpoints are crucial for maintaining self-tolerance to prevent autoimmune disease and protect against tissue inflammation and damage [13]. Conversely, excessive expression of immune checkpoint proteins has been associated with immune resistance used by tumor cells to escape from antitumoral immunity [14]. Recent advances in cancer immunotherapy have highlighted the importance of PD-1/PD-L1 immune checkpoint blockade to suppress antigen-reactive TCR signaling with high efficacy in patients with advanced solid tumors [15,16,17]. PD-1 is expressed on activated T and B cells and its engagement by its ligand PD-L1 disrupts TCR activation and downstream signaling. Resulting immunosuppression involves several mechanisms, including T-cell apoptosis, T-cell exhaustion, T-cell anergy, IL-10 production and the induction of regulatory T cells [11]. In contrast with immune resistance, exacerbated immunity leads to immune-mediated tissue injury and autoimmune disease. PD-1 and PD-L1 deficiency have been associated with a lupus-like syndrome phenotype in mice [18]. Moreover, lack of PD-1 or PD-L1 accelerates experimental diabetes and autoimmune encephalitis [19,20]. On a mechanistic level, PD-L1 depletion in antigen-presenting cells results in a failure to convert naïve CD4^+^ into regulatory T cells (22) and PD-1 knockout mice are prone to enrich for Th1 and Th17 cells [21,22]. The fact that specifically PD-1 inhibitors can cause ANCA-associated renal vasculitis implicates a comparable mechanistic link between impaired immune checkpoint PD-1 signaling (by either targeted PD-1 therapy or loss of PD-1 itself) and disease activation [3]. Finally, PD-1 was associated with decreased local synthesis of complement factor B. A causal role of complement factor B has already been shown since its deficiency completely inhibited the development of renal vasculitis in mice [9]. Interestingly, we did not observe a correlation between PD-1 and complement C5 or its C5a receptor. Combined with our observations, this may imply a link between impaired PD-1/PD-L1 signaling, complement factor B and active ANCA-associated renal vasculitis. These findings could be of relevance because experimental data have already described that PD-1 agonism can be used therapeutically to attenuate autoimmunity in multiple disease models [23]. Furthermore, targeted therapy against both complement C5/C5a receptor and factor B is available and currently evolving in the treatment of AAV. Therefore, this pilot study expands our current knowledge and describes a potential interplay between immune checkpoints and the alternative complement pathway in active ANCA-associated renal vasculitis.

Our study has several limitations, such as the small patient number and the retrospective study design. Furthermore, our observations are associative and do not prove causality, requiring mechanistic studies including protein levels of complement pathway components and in vitro culture models. Nevertheless, we hereby provide evidence for a potential interplay between immune checkpoints and the alternative complement pathway in active ANCA-associated renal vasculitis. Moreover, these findings might also be expandable and of interest for other inflammatory renal diseases.

## 4. Materials and Methods

### 4.1. Study Population and Approval

A total of 15 kidney biopsies with ANCA-associated renal vasculitis at the University Medical Center Göttingen were included (Appendix A); part of the patient cohort has previously been described [24]. At the time of kidney biopsy, all patients received steroids, and further remission induction therapy was initiated thereafter based on histopathological confirmation of ANCA-associated renal vasculitis. Steroids were administered either as intravenous pulse therapy or orally with a tapering schedule. The use of parts of human specimens for research purposes was approved by the Ethics Committee of the University Medical Center Göttingen (protocol code: 28/09/17, approval date 17 November 2017). All patients gave written informed consent for the use of routinely collected data for research purposes as part of their regular medical care in the contract of the University Medical Center Göttingen, and all samples were deidentified.

### 4.2. Renal Histopathology

A kidney pathologist evaluated the kidney biopsies, and these were blinded to clinical data. Within a kidney biopsy, each glomerulus was scored separately for the presence of necrosis, crescents and global sclerosis. Based on these scorings, histopathological subgrouping according to Berden et al. was performed [25]. Furthermore, the ANCA renal risk score (ARRS) was evaluated according to Brix et al. [26]. Tubulointerstitial lesions were also evaluated analogously to the Banff scoring system for allograft pathology, as described previously [27]. Generally, Banff score lesions include interstitial inflammation (*i*), tubulitis (*t*), interstitial fibrosis (*ci*), tubular atrophy (*ct*), total inflammation (*ti*), inflammation in areas of IFTA (*i-IFTA*) and tubulitis in areas of IFTA (*t-IFTA*) [27]. The Banff scoring system had three grades: none (0), mild (1), moderate (2) and severe (3). The cut-off points for *i* were <10%, 10–25%, 26–50% and >50%, respectively. Cut-off points for *t* were 0, 1–4, 5–10 and >10 mononuclear cells/tubular cross section. Cut-off points for *v* were no arteritis, mild-to-moderate intimal arteritis in at least 1 arterial cross section, severe intimal arteritis with at least 25% luminal area lost in at least 1 arterial cross section, transmural arteritis and/or arterial fibrinoid change and medial smooth muscle necrosis with lymphocytic infiltration in vessel, respectively. Cut-off points for *g* were no glomerulitis, segmental or global glomerulitis in less than 25% of glomeruli, segmental or global glomerulitis in 25 to 75% of glomeruli and segmental or global glomerulitis in more than 75% of glomeruli. Cut-off points for *ci* were interstitial fibrosis in up to 5%, 6–25%, 26–50% and >50% of the cortical area. Cut-off points for *ct* were no tubular atrophy and tubular atrophy involving up to 25%, 26–50% and >50% of the area of cortical tubules. Cut-off points for *ah* were no PAS-positive hyaline arteriolar thickening, mild-to-moderate PAS-positive hyaline thickening in at least 1 arteriole, in more than 1 arteriole and in many arterioles. Cut-off points for *ptc* were a maximum number of leukocytes <3, at least 1 leukocyte cell in ≥10% of cortical peritubular capillaries (PTCs) with 3–4 leukocytes in most severely involved PTC, at least 1 leukocyte in ≥10% of cortical PTC with 5–10 leukocytes in most severely involved PTC and at least 1 leukocyte in ≥10% of cortical PTC with >10 leukocytes in most severely involved PTC. Cut-off points for *ti* were <10%, 10–25%, 26–50% and >50% of total cortical parenchyma inflamed. Cut-off points for *i-IFTA* and *t-IFTA* were no inflammation or less than 10%, 10–25%, 26–50% and >50% of scarred cortical parenchyma.

### 4.3. Immunostaining

Immunostainings were performed on 2 μm formalin-fixed, paraffin-embedded kidney sections. Kidney sections were deparaffinized in xylene and rehydrated in ethanol containing distilled water. Tissue sections were pre-treated using proteinase K antigen retrieval (DAKO, Glostrup, Denmark) and stained using primary antibodies against PD-1 (1:500, ab52587, Abcam, Cambridge, UK) and PD-L1 (1:100, ab205921, Abcam, Cambridge, UK); labeling was performed using Novolink^TM^ Polymer Detection System (Leica Biosystems, Wetzlar, Germany) according to the manufacturer’s protocol. Nuclear counterstain was performed by using Mayer’s hematoxylin solution (Sigma, St. Louis, MO, USA). As described previously, interstitial cells positive for PD-1 were evaluated using mean values of 10 randomly selected cortical visual fields at 400× magnification and scored semiquantitatively (0: no cell per visual field, 1: 1–3 cells per visual field, 2: 3–6 cells per visual field, 3: >6 cells per visual field) [10]. The intensity of PD-L1 staining was evaluated at 400× magnification and scored semiquantitatively (0: no staining, 1: weak and segmental staining, 2: moderate staining, 3: strong staining) [10].

### 4.4. Analyses of Publicly Available Array Datasets

Publicly available datasets were analyzed for PD-1 expression (encoded by *PDCD1*) from Nephroseq (www.nephroseq.org, 30 September 2022, University of Michigan, Ann Arbor, MI). Particularly, median-centered log_2_ *PDCD1* mRNA expression levels (reporter ID: 5133, platform: Affymetrix Human Genome U133 Plus 2.0 Array, altCDF v10) were extracted specifically from microdissected glomerular (21 with renal vasculitis, Appendix A) and tubulointerstitial compartments (31 healthy controls, 21 with renal vasculitis, Appendix A) [28]. In addition, tubulointerstitial mRNA expression of *C1QA* (reporter ID: 712), *C1QB* (713), *C2* (717), *C3* (718), *C3AR1* (719), *C5* (727), *C5AR1* (728), *CFB* (629), *CFD* (1675), *CFH* (3075), *CFP* (5199), *CR1* (1378) and *CR2* (1380) were extracted (31 healthy controls, 21 with renal vasculitis) [28].

### 4.5. Statistical Methods

Variables were tested for normal distribution using the Shapiro–Wilk test. Single-group comparison was performed using unpaired Student’s *t* test. Pearson’s (for normal distribution) and Spearman’s correlations (for non-normal distribution) were used to assess the univariate correlation between continuous variables. Multivariate analysis was performed by stepwise linear regression. Data analyses were performed with GraphPad Prism (version 9.3.1 for macOS, GraphPad Software, San Diego, CA, USA) and IBM SPSS Statistics (version 27 for MacOS, IBM Corporation, Armonk, NY, USA). A value of *p* < 0.05 was considered statistically significant.

## 5. Conclusions

We hereby provide evidence for a potential interplay between immune checkpoints and the alternative complement pathway in active ANCA-associated renal vasculitis. These observations might contribute to a better understanding of pathomechanisms underlying complement system activation and dysregulation in ANCA-associated renal vasculitis.

## Figures and Tables

**Figure 1 ijms-24-02975-f001:**
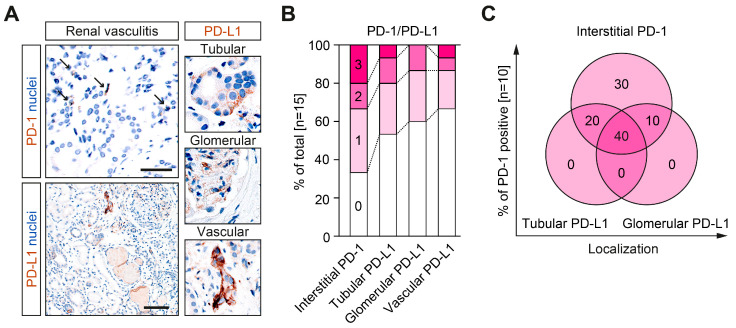
Immune checkpoints PD-1 and PD-L1 are present in different renal compartments in ANCA-associated renal vasculitis. (**A**) Immunostainings for PD-1 (scale bar: 40 μm) and PD-L1 (scale bar: 100 μm) are shown; counterstaining was performed by using hematoxylin. Presence of PD-1 was limited to interstitial cell positivity (arrows); PD-L1 was found in the tubular, glomerular and vascular compartment in ANCA-associated renal vasculitis. (**B**) Quantification of PD-1 and PD-L1 is shown as fraction of total (n = 15). (**C**) Among cases with interstitial PD-1 (n = 10), frequency of tubular and glomerular PD-L1 positivity is shown. In case of PD-1 positivity, most were also positive for tubular and/or glomerular PD-L1.

**Figure 2 ijms-24-02975-f002:**
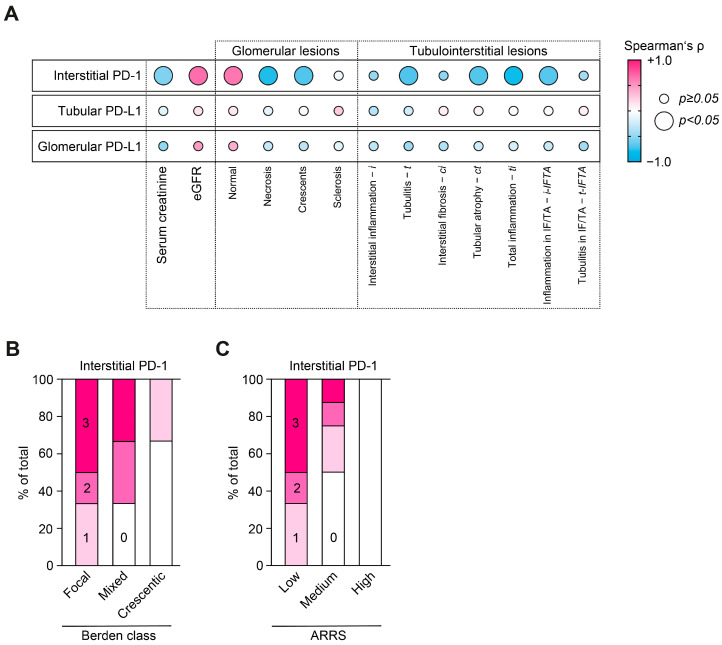
Loss of interstitial PD-1 correlates with active ANCA-associated renal vasculitis. (**A**) Correlations between PD-1 and PD-L1 positivity, markers of kidney injury and histopathological lesions in ANCA-associated renal vasculitis (n = 15) are shown by heatmap reflecting mean values of Spearman’s *ρ*; circle size represents significance level. (**B**,**C**) Quantification of interstitial PD-1 is shown as fraction of total according to Berden classification and ARRS.

**Figure 3 ijms-24-02975-f003:**
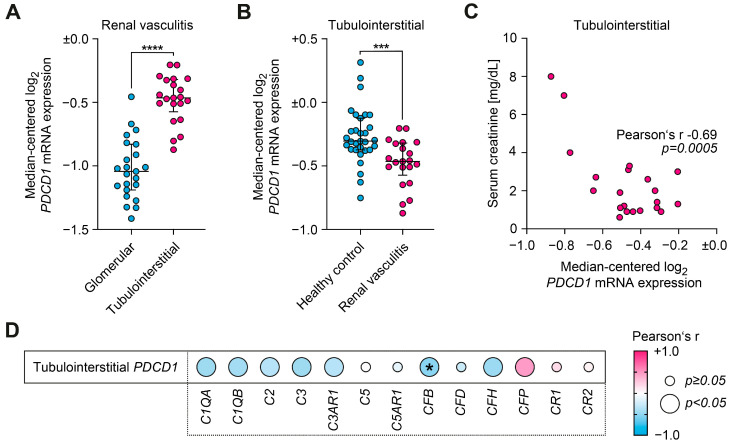
PD-1 associates with decreased local synthesis of complement factor B. (**A**) Direct comparison between glomerular (n = 23) and tubulointerstitial median-centered log_2_ *PDCD1* mRNA expression (n = 21) in renal vasculitis; comparison of groups was performed using unpaired *t* test (**** *p* < 0.0001). *PDCD1* mRNA expression was predominantly found within the tubulointerstitial as compared to the glomerular compartment. (**B**) Direct comparison of tubulointerstitial median-centered log_2_ *PDCD1* mRNA expression between healthy controls (n = 31) and renal vasculitis (n = 21); comparison of groups was performed using unpaired *t* test (*** *p* < 0.001). *PDCD1* mRNA expression was significantly lower in ANCA-associated renal vasculitis as compared to healthy control kidneys. (**C**) Correlations between tubulointerstitial median-centered log_2_ *PDCD1* mRNA expression and serum creatinine levels (n = 21) with Pearson’s r and significance level are shown. Loss of intrarenal *PDCD1* mRNA expression correlated with increased serum creatinine measurements. (**D**) Correlations between tubulointerstitial median-centered log_2_ mRNA expression levels of *PDCD1* and various complement components including healthy control kidneys (n = 31) and ANCA-associated renal vasculitis (n = 21) are shown by heatmap reflecting mean values of Pearson’s r. Circle size represents significance level in the univariate analysis, asterisk (*) in the multivariate analysis. *PDCD1* mRNA expression associated with decreased expression levels of *CFB*.

## Data Availability

The original contributions presented in the study are included in the article/Appendix A; further data and material are available from the corresponding author upon reasonable request.

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
