# Peer review of "Association between Loss of Immune Checkpoint Programmed Cell Death Protein 1 and Active ANCA-Associated Renal Vasculitis"

_ijms, 2023, doi:10.3390/ijms24032975_

Round 1
Reviewer 1 Report
The study of Hakroush et al reports on a possible relation between intrarenal PD-1 and PD-L1 expression and (active) AAV with renal involvement. This stud was performed on 15 kidney biopsies with ANCA-associated renal vasculitis and publicly available datasets. In essence they found that PD1 expression was predominantly found in the tubulointerstitium and was decreased in AAV. They showed that interstitial PD-1 correlated with tubular and/or glomerular PD-L1 positivity and that it negatively correlated with complement components of which only factor B remained significant in multivariate analysis.
Although this is a highly interesting study there remains a lot of questions unanswered.
1) PD-1 is mainly expressed on activated T cells, B cells, and monocyte, and it seems from Figure 2A that los of interstitial PD1 expression correlates with more typical inflammatory histological scores e.g. Tubulitis, total inflammation and inflammation in IFTA. It would therefore be intriguing to know if activated cells in these lesions don’t express PD1 or alternatively that cells in these lesions are not activated.
2) I was also wondering if loss of PD1 only occurs in active renal lesions? Was it only observed in AAV? If not how was the relation with complement expression in other inflammatory renal disease entities.
3) It is not clear why the authors did not confirm decreased complement production in their 15 kidney biopsies by means of staining. Where exactly was complement production observed? Was there a general decrease or only in tubular compartments?
4) The paper would have gained more significance if PD1-PDL1 signaling in relation to complement production using in vitro culture models (tubular cells mesangial cells) was included.
Minor:
The paper is written in a very concise manner, and gives the impression that it is intended as a brief communication. If so, I think that the abstract might be shortened, if not I think that the discussion might favor from the author’s view on how PD1-PDL1 signaling might affect local complement production in the kidney.
Author Response
The study of Hakroush et al reports on a possible relation between intrarenal PD-1 and PD-L1 expression and (active) AAV with renal involvement. This stud was performed on 15 kidney biopsies with ANCA-associated renal vasculitis and publicly available datasets. In essence they found that PD1 expression was predominantly found in the tubulointerstitium and was decreased in AAV. They showed that interstitial PD-1 correlated with tubular and/or glomerular PD-L1 positivity and that it negatively correlated with complement components of which only factor B remained significant in multivariate analysis.
Although this is a highly interesting study there remains a lot of questions unanswered.
We thank the referee for her/his time to evaluate our study. Based on your valuable comments, we now provide a revised version of the manuscript. Please find a point-to-point response to your comments below.
1) PD-1 is mainly expressed on activated T cells, B cells, and monocyte, and it seems from Figure 2A that los of interstitial PD1 expression correlates with more typical inflammatory histological scores e.g. Tubulitis, total inflammation and inflammation in IFTA. It would therefore be intriguing to know if activated cells in these lesions don’t express PD1 or alternatively that cells in these lesions are not activated.
Thank you for this important comment! We now provide an in-depth discussion about possible mechanisms underlying our observations:
Experimental studies have already described that disruption of the tissue-protective PD-1/PD-L1 checkpoint unleashes immunity in the pathogenesis of medium and large vessel vasculitites [6]. T-cell–dependent immune responses are modulated by costimu-latory and coinhibitory stimuli, particularly receptor-ligand interactions that modulate T-cell receptor (TCR) signaling [7]. Such immune checkpoints are crucial for maintaining self-tolerance to prevent autoimmune disease, and protect against tissue inflammation and damage [8]. Conversely, excessive expression of immune checkpoint proteins has been associated with immune resistance used by tumor cells to escape from antitumoral immunity [9]. Recent advances in cancer immunotherapy have highlighted the im-portance of PD-1/PD-L1 immune checkpoint blockade to suppress antigen-reactive TCR signaling with high efficacy in patients with advanced solid tumors [10-12]. PD-1 is ex-pressed on activated T and B cells and its engagement by its ligand PD-L1 disrupts TCR activation and downstream signaling. Resulting immunosuppression involves several mechanisms, including T-cell apoptosis, T-cell exhaustion, T-cell anergy, IL-10 produc-tion, and induction of regulatory T cells [6]. In contrast to immune resistance, exacerbated immunity leads to immune-mediated tissue injury and autoimmune disease. PD-1 and PD-L1 deficiency have been associated with a lupus-like syndrome phenotype in mice [13]. Moreover, lack of PD-1 or PD-L1 accelerates experimental diabetes and autoimmune encephalitis [14, 15]. On a mechanistic level, PD-L1 depletion in antigen-presenting cells results in failure to convert naïve CD4+ into regulatory T cells (22), and PD-1 knockout mice are prone to enrich for Th1 and Th17 cells [16, 17]. The fact that specifically PD-1 inhibitors can cause ANCA-associated renal vasculitis implicates a comparable mecha-nistic link between impaired immune checkpoint PD-1 signaling (by either targeted PD-1 therapy or loss of PD-1 itself) and disease activation [3].
2) I was also wondering if loss of PD1 only occurs in active renal lesions? Was it only observed in AAV? If not how was the relation with complement expression in other inflammatory renal disease entities.
This is a very interesting point, thank you! However, we here focused on ANCA-associated renal vasculitis since recent reports implicated that specifically PD-1 inhibitors could cause de novo or relapsing ANCA-associated renal vasculitis (Aqeel F et al. RMD Open 2022). Therefore, we now resubmit the revised manuscript as Brief Report and discussed that these findings might also be expandable and of interest for other inflammatory renal diseases.
3) It is not clear why the authors did not confirm decreased complement production in their 15 kidney biopsies by means of staining. Where exactly was complement production observed? Was there a general decrease or only in tubular compartments?
We agree that confirmatory imunostainings would strengthen the overall conclusion. We here provide evidence that PD-1 was associated with decreased local synthesis of complement factor B specifically in the tubulointerstitial compartment based on transcriptomic data. Because these observations require confirmation, we now submit our study as Brief Report and clearly state that limitation in the revised version of the manuscript.
4) The paper would have gained more significance if PD1-PDL1 signaling in relation to complement production using in vitro culture models (tubular cells mesangial cells) was included.
We completely agree with the referee that in vitro culture models could improve our mechanistic understanding. Due to the preliminary nature of our observations, we now submit our study as Brief Report and clearly state that limitation in the revised version of the manuscript.
Minor:
The paper is written in a very concise manner, and gives the impression that it is intended as a brief communication. If so, I think that the abstract might be shortened, if not I think that the discussion might favor from the author’s view on how PD1-PDL1 signaling might affect local complement production in the kidney.
We agree with the referee and resubmit the revised version of the manuscript as Brief Report.
Reviewer 2 Report
The introduction and the discussion sections of the manuscript titled “Association between loss of immune checkpoint programmed cell death protein 1 and active ANCA-associated renal vasculitis” need extend. The followings are some concerns and comments have been pointed out that the authors may want to consider.
1) Line 40: The introduction section needs extend. More information is required.
2) Lines 103-109: This part should be “Figure 3” instead of “Figure 1”.
3) Line 111 Figure 3: a) How can we understand the relationship between different samples? (PDCD1 mRNA in tubulointerstitial and serum creatinine). b) Protein expression might more important than the mRNA level. Why did the authors only mention and test PDCD1 mRNA expression?
4) Line 117: It should be “student t-test”.
5) Line 127: The discussion section needs extend.
6) Line 155 Materials and Methods section: More information is needed to allow other researchers to reproduce your work relatively easier.
7) Is there any in-depth investigation instead of the current preliminary data?
Author Response
The introduction and the discussion sections of the manuscript titled “Association between loss of immune checkpoint programmed cell death protein 1 and active ANCA-associated renal vasculitis” need extend. The followings are some concerns and comments have been pointed out that the authors may want to consider.
We thank the referee for her/his time to evaluate our study. Based on your valuable comments, we now provide a revised version of the manuscript. Please find a point-to-point response to your comments below.
1) Line 40: The introduction section needs extend. More information is required.
We agree that the introduction section required extension, therefore we expanded the background information:
AAV is a small vessel vasculitis, most frequently presenting as microscopic polyangiitis (MPA) or granulomatosis with polyangiitis (GPA) [4, 5]. Acute kidney injury (AKI) is a common and severe complication of AAV as it can cause progressive chronic kidney disease (CKD), end-stage renal disease (ESRD) or death [6, 7]. Pathogenic ANCAs, in particular proteinase 3 (PR3) and myeloperoxidase (MPO), trigger a deleterious immune response resulting in pauci-immune necrotizing and crescentic renal vasculitis, a common manifestation of glomerular injury in AAV [8]. Seminal work has shown that complement depletion by cobra venom factor and blockade of the alternative complement pathway protected from experimental renal vasculitis [9]. Contrasting to this, complement C4 knockdown as shared component of the classical and lectin pathway had no effect [9]. These findings first demonstrated that the alternative complement pathway is a pre-requisite of ANCA-induced lesions. Interestingly, interstitial PD-1 has already been described as determinant of susceptibility for kidney injury in the context of ICIs and has been linked to regulation by the complement system [10].
2) Lines 103-109: This part should be “Figure 3” instead of “Figure 1”.
Thank you, the paragraph was corrected accordingly.
3) Line 111 Figure 3: a) How can we understand the relationship between different samples? (PDCD1 mRNA in tubulointerstitial and serum creatinine).
Correlations between tubulointerstitial median-centered log2 PDCD1 mRNA expression and serum creatinine levels with Pearson’s r and significance level are shown in Figure 3C. Since both parameters are continuous variables, this comparison is reasonable and confirmed the association between loss of intrarenal PDCD1 mRNA expression and increased serum creatinine measurements.
b) Protein expression might more important than the mRNA level. Why did the authors only mention and test PDCD1 mRNA expression?
We agree that confirmatory imunostainings would strengthen the overall conclusion. We here provide evidence that PD-1 was associated with decreased local synthesis of complement factor B specifically in the tubulointerstitial compartment based on transcriptomic data. Because these observations require confirmation, we now submit our study as Brief Report and clearly state that limitation in the revised version of the manuscript.
4) Line 117: It should be “student t-test”.
Thank you, the method section was corrected accordingly.
5) Line 127: The discussion section needs extend.
Thank you for pointing this out as also mentioned by referee 1. We now provide an in-depth discussion about possible mechanisms underlying our observations:
Experimental studies have already described that disruption of the tissue-protective PD-1/PD-L1 checkpoint unleashes immunity in the pathogenesis of medium and large vessel vasculitites [6]. T-cell–dependent immune responses are modulated by costimu-latory and coinhibitory stimuli, particularly receptor-ligand interactions that modulate T-cell receptor (TCR) signaling [7]. Such immune checkpoints are crucial for maintaining self-tolerance to prevent autoimmune disease, and protect against tissue inflammation and damage [8]. Conversely, excessive expression of immune checkpoint proteins has been associated with immune resistance used by tumor cells to escape from antitumoral immunity [9]. Recent advances in cancer immunotherapy have highlighted the im-portance of PD-1/PD-L1 immune checkpoint blockade to suppress antigen-reactive TCR signaling with high efficacy in patients with advanced solid tumors [10-12]. PD-1 is ex-pressed on activated T and B cells and its engagement by its ligand PD-L1 disrupts TCR activation and downstream signaling. Resulting immunosuppression involves several mechanisms, including T-cell apoptosis, T-cell exhaustion, T-cell anergy, IL-10 produc-tion, and induction of regulatory T cells [6]. In contrast to immune resistance, exacerbated immunity leads to immune-mediated tissue injury and autoimmune disease. PD-1 and PD-L1 deficiency have been associated with a lupus-like syndrome phenotype in mice [13]. Moreover, lack of PD-1 or PD-L1 accelerates experimental diabetes and autoimmune encephalitis [14, 15]. On a mechanistic level, PD-L1 depletion in antigen-presenting cells results in failure to convert naïve CD4+ into regulatory T cells (22), and PD-1 knockout mice are prone to enrich for Th1 and Th17 cells [16, 17]. The fact that specifically PD-1 inhibitors can cause ANCA-associated renal vasculitis implicates a comparable mecha-nistic link between impaired immune checkpoint PD-1 signaling (by either targeted PD-1 therapy or loss of PD-1 itself) and disease activation [3].
6) Line 155 Materials and Methods section: More information is needed to allow other researchers to reproduce your work relatively easier.
We apologize for the missing information and revised the method section accordingly with a detailed description of histopathological analysis, immunostaining and array data analysis. We belief that this improved reproducibility of our findings.
7) Is there any in-depth investigation instead of the current preliminary data?
We completely agree with the referee that an in-depth investigation (including in vitro culture models) could improve our mechanistic understanding. Due to the preliminary nature of our observations, we now submit our study as Brief Report and clearly state that limitation in the revised version of the manuscript.
Round 2
Reviewer 2 Report
Thank you for the update. It is good to be a brief report. Good luck.